# The Interaction between Gut Microbiota and Host Amino Acids Metabolism in Multiple Myeloma

**DOI:** 10.3390/cancers15071942

**Published:** 2023-03-23

**Authors:** Qin Yang, Yumou Wei, Yinghong Zhu, Jiaojiao Guo, Jingyu Zhang, Yanjuan He, Xin Li, Jing Liu, Wen Zhou

**Affiliations:** 1Haihe Laboratory of Cell Ecosystem, State Key Laboratory of Experimental Hematology, National Clinical Research Center for Geriatric Disorders, Department of Hematology, Xiangya Hospital, Central South University, Changsha 410008, China; 2Key Laboratory for Carcinogenesis and Invasion, Chinese Ministry of Education, Key Laboratory of Carcinogenesis, Chinese Ministry of Health, Cancer Research Institute, School of Basic Medical Sciences, Central South University, Changsha 410008, China; 3Department of Hematology, The Third Xiangya Hospital, Central South University, Changsha 410013, China

**Keywords:** multiple myeloma, gut microbiota, host metabolism, amino acid metabolism, gut-microbiota–host-metabolic interaction

## Abstract

**Simple Summary:**

Multiple myeloma (MM) is the second most common hematological malignancy and remains incurable. Recent evidence substantiates the interaction of gut microbiota and MM, together with abnormal amino acid metabolism and MM. Moreover, the association between gut microbiota and host amino acid metabolism on MM has been highlighted. This article presents a review of the literature on the relationship between gut microbiota, metabolism, and MM, together with strategies to modulate the microbiota.

**Abstract:**

Although novel therapies have dramatically improved outcomes for multiple myeloma (MM) patients, relapse is inevitable and overall outcomes are heterogeneous. The gut microbiota is becoming increasingly recognized for its influence on host metabolism. To date, evidence has suggested that the gut microbiota contributes to MM, not only via the progressive activities of specific bacteria but also through the influence of the microbiota on host metabolism. Importantly, the abnormal amino acid metabolism, as well as the altered microbiome in MM, is becoming increasingly apparent, as is the influence on MM progression and the therapeutic response. Moreover, the gut-microbiota–host-amino-acid metabolism interaction in the progression of MM has been highlighted. Modulation of the gut microbiota (such as fecal microbiota transplantation, FMT) can be modified, representing a new angle in MM treatment that can improve outcomes. In this review, the relationship between gut microbiota, metabolism, and MM, together with strategies to modulate the microbiota, will be discussed, and some unanswered questions for ongoing and future research will be presented.

## 1. Introduction

Multiple myeloma (MM) is a hematologic malignancy with abnormal proliferation of clonal plasma cells in the bone marrow, which accounts for the second most common hematological malignancy [1] and is characterized by the manifestations of destructive bone lesions, kidney injury, anemia, and hypercalcemia [2]. Tremendous progress has been made in both the pathogenesis and treatment of MM, thereby largely improving clinical outcomes with a median survival of 5–7 years [3]. Nevertheless, the inevitable occurrence of treatment resistance and relapse seriously threatens patients’ lives, and heterogeneous overall outcomes remain challenging for researchers [1], which requires us to explore new angles for a better understanding of MM. Recent evidence substantiates the association between gut microbiota and diseases, especially cancer. However, whether gut microbiota influences MM remains to be explored.

The human gut microbiota is comprised of trillions of bacteria and other microorganisms that live on and inside humans [4], most of which belong to the phyla of *Firmicutes*, *Bacteroidetes*, *Actinobacteria,* and *Proteobacteria* [5]. The composition of the gut microbiota is shaped by several factors (including host genetics, colonization at the time of birth, type of birth delivery, lifestyle, and exposure to antibiotics [6,7,8]), which remains relatively constant throughout adult life except for the influence of diet, change of lifestyle, diseases, and corresponding treatment [9]. As the understanding of the microbiota grows, it is becoming increasingly apparent that the gut microbiota plays a critical role in human health, highlighted by the functions of maintaining local barrier homeostasis to regulating systemic metabolism and immunity [4,10]. The disruption of the gut microbiota (dysbiosis) could be associated with a range of diseases by affecting gut homeostasis, systemic metabolism, and immunity [4].

Recently, emerging evidence has highlighted the crucial crosstalk between the gut microbiota and cancer, not only from the carcinogenesis and development but also the therapeutic response and susceptibility to toxicities, which is defined by the phenomenon from which the pro-carcinogenic phenotype of genetically mutated mice has been revealed to be transferable to wild-type mice via gut microbiota [11,12]. Gut microbiota has also been reported to be involved in the progression of varieties of cancers [13,14,15], and the transfer of the fecal microbiota from patients with higher response to cancer therapy into germ-free mice has been demonstrated to efficiently increase the response to the therapy in those mice [16,17]. Thus, it can be speculated that the changes in gut microbiota may directly or indirectly affect carcinogenesis and development of cancer, together with a response to cancer therapy. In line with this, the involvement of gut microbiota and hematologic cancers and the therapy of hematopoietic stem cell transplantation (HSCT) has been explored, defining the crucial association of gut microbiota and hematologic cancers, especially MM, probably through modification of the host metabolism.

After the first report of the metabolic alterations of cancer cells [18], the field of cancer metabolism has become a topic of renewed interest in the past decade. Aided by new biochemical and molecular biological tools, studies in cancer metabolism have expanded our understanding of the mechanisms and functional consequences of metabolic alterations on cancer [19,20], emphasizing the association between metabolic reprogramming and the development and treatment of cancer [20,21,22]. Furthermore, based on the ability of gut microbiota to modulate host metabolism, studies have largely focused on the role of gut microbiota-host metabolism interaction in cancer [23,24]. MM, which exists in the malignant bone marrow microenvironment, has been reported to represent unique metabolism characteristics (particularly amino acid metabolism) [25], along with alterations of gut microbiota [26]. How and why they appear necessitates an overall understanding of microbial metabolism (amino acid) on the host, metabolic reprogramming, and alterations of gut microbiota of MM, as well as the impact of their interaction on MM. Therefore, the alteration of metabolism and microbiome, along with their interaction with MM, will be discussed herein.

## 2. Gut Microbiota and Amino Acid Catabolism

Undigested dietary components are fermented by the anaerobic microbial community and produce a wide range of metabolites and then affecting the host [27]. The gut microbe can exhibit one of two strategies in the initial stage of amino acid catabolism: either deamination to carboxylic acid and ammonia or decarboxylation to amine and carbon dioxide [28]. As one of the end-products (Figure 1), ammonia has been shown to inhibit mitochondrial oxygen consumption and reduce short-chain fatty acids (SCFAs) catabolism, leading to the assumptive negative impact of excess ammonia on the host [29]. Moreover, a relatively low concentration of ammonia has been confirmed to increase mucosal damage and further cause colorectal adenocarcinoma, showing potential carcinogenic effects [30]. However, the concentration of ammonia could be regulated by the host to reduce its toxicity, probably by assimilation into microbe for microbial amino acid biosynthetic processes, or through conversion to citrulline and glutamine in host intestinal epithelial cells, together with slowing the release into the bloodstream [31,32]. Furthermore, SCFAs can be produced by the function of amino acid fermentation. The ketogenic and glucogenic metabolic pathways are processed to produce SCFAs (butyrate and propionate) (Figure 1), and then utilized for the metabolic needs of the colon and the body [33,34]. Several studies have shown the remarkable interactions between SCFAs and anti-inflammatory and anti-apoptotic effects on cancer [35]. The recent studies mainly aimed at the end-products of amino acids, while the role of secondary metabolites on MM during the metabolic process of amino acids is still unclear. A deeper understanding of this topic in the future may show this to be a potential mechanism and modulation in MM.

The fermentation of amino acids could produce a variety of potentially bioactive products by several *Bacteroides* spp. and some *Firmicutes* [30,36,37,38]. Some nitrogenous products, especially N-nitroso compounds (NOCs), have emphasized a significant fecal increase in high-protein dietary individuals [38]. Moreover, the increase in the dietary intake of NOCs has been reported to be positively correlated with colorectal cancer [39], and exert carcinogenic effects potentially via DNA alkylation [40]. In addition, polyamines synthesis also occurs in gut bacteria [41], together with up-regulated production by host cells promoted by certain gut bacteria, such as enterotoxigenic *Bacteroides fragilis* [42]. Polyamines are involved in varieties of physiological functions [35]. However, high levels of polyamines have been reported to be associated with cancer, probably by oxidative stress that results from polyamine catabolism [42]. In addition, some bacteria exhibit toxicity through polyamines [43]. 

## 3. Abnormal Amino Acid Metabolism and Multiple Myeloma

Given this growing development of the metabolic function of gut microbiota, it is becoming increasingly clear that gut microbiota participates in host metabolic processes and exerts influences on the carcinogenesis and development of cancer. Studies to further dissect the gut microbiota–metabolic interactions underlying the myelomagenesis, progression, and anti-myeloma responses were discussed. Among them, myeloma-related metabolic reprogramming has been defined.

### 3.1. The Effects of Abnormal Amino Acid Metabolism on MM

Different from other solid tumor cells, MM cells produce abundant ineffective monoclonal immunoglobulin. In line with this, different metabolic characteristics may exist in MM. Studies have provided strong evidence for this speculation. Aided by high-performance liquid chromatography (HPLC) and mass spectrometry detection, metabonomics analysis of MM patients and healthy individuals was performed and demonstrated that the differential metabolites were mainly enriched in amino acid metabolism [44]. Additional cohorts have also been studied by untargeted/targeted metabolomics, showing the abnormal amino acid metabolisms in MM, rather than the traditional views of abnormal glucose metabolism in a majority of cancers, which may be related to the biological function of MM [25,44,45,46,47,48]. Therefore, differential amino acid metabolic profiles in MM patients have been brought to this topic and are summarized based on the insights gained from the results.

Accounting for the onset of MM, the stage of MGUS has been studied to explore myelomagenesis. A substantially different metabolomic profile of MGUS and MM patients and healthy individuals was determined, which is characterized by a significant difference in glutamate between MM and MGUS, with the potential involvement of amino acid metabolism disorders in the myelomagenesis [48]. The results were further supplemented by multiple studies demonstrating the role of amino acids in MM patients. Our inspirational findings involved substantiating the higher concentrations of glycine, serine, proline, and glutamate in MM patients [25]. Differential metabolites of leucine, tryptophan, and valine have also been drawn in MM patients; these metabolites are considered evidence for the pathogenesis and development of MM [44]. In addition, a disorder of glutamate metabolism in the bone marrow microenvironment of MM patients has been highlighted, demonstrating increased glutamate and decreased glutamine [49]. The group also studied and defined serum aspartic acid as a candidate biomarker for diagnosis, while serum threonine was for risk prediction [49].

### 3.2. The Effects of Abnormal Amino Acid Metabolism on MM Progression

Importantly, the impacts of the amino acids on MM progression have also been studied in the setting of association with MM tumor burden. Our findings involved substantiating the accelerative function, demonstrating that the MM group with high glycine concentration in the bone marrow had a more serious ISS stage and higher plasma cell percentages. Our further investigations into mechanisms confirmed that glycine contributed to MM cell proliferation through glutathione synthesis. Effectively blocking the utilization of glycine may inhibit proliferation, suggesting that targeted amino acid metabolism (such as glycine metabolism) may have therapeutic potential in MM [25]. Another study assessed serine synthesis on MM progression, demonstrating that the increase of serine synthesis was related to the proliferation of MM cells, and higher expression of the related gene PHGDH was associated with a poor prognosis of MM [50]. Furthermore, abnormal amino acid metabolism promoted MM progression through the immunosuppressive microenvironment [51].

In addition to these studies on MM progression, the impact of abnormal amino acids has been studied in myeloma-related end-organ dysfunction. As reported, higher glycine in the bone marrow was significantly correlated with bone destruction in MM patients. Conversely, osteolysis had been alleviated in MM mice by glycine-free diets [25]. Our group also proposed the notion of a positive correlation between higher glutamine and pneumonia in MM. This has demonstrated the aggravative function of increased inflammatory factors by glutamine [52]. Given the importance of abnormal amino acids in MM progression, it is possible that targeting them could serve as a potential strategy to improve clinical outcomes. However, efforts to identify ideal targets remain to be defined.

### 3.3. The Effects of Abnormal Amino Acid Metabolism on MM Therapy

Based on the available literature, it is clear that amino acid metabolism is associated with drug response and toxicity of MM. To be specific, glycine [25], as well as proline [53], has demonstrated an increase in the resistance of Bortezomib (BTZ). Furthermore, inhibition of glycine utilization and PHGDH inhibitors respectively may enhance the effects of BTZ on MM cells. Moreover, inhibition of the L-glutamine transporter ASCT2 has been reported to enhance the sensitivity of MM cells to proteasome inhibitors [54]. In line with this, amino acid depletion triggered by L-asparaginase has been reported to sensitize MM cells to carfilzomib by inducing mitochondria ROS-mediated cell death [55]. Furthermore, targeting MM glutamine metabolism has been confirmed to enhance the binding of the BIM gene to BCL-2, thereby increasing the sensitivity of MM cells to Venetoclax, providing an effective treatment strategy for relapsed refractory MM [56].

Overall, these data provide evidence that the metabolic pattern in MM is mainly characterized by abnormal amino acid metabolism, especially abnormal glycine and serine metabolism. The impact of the abnormal amino acids on MM progression and the therapeutic response has also been defined. Thus, it is increasingly clear that targeted amino acid metabolism may represent a novel and important adjunct to current anti-myeloma therapeutic modalities. However, major challenges remain. Furthermore, efforts to identify the origin of abnormal amino acid metabolism of MM to better understand MM are underway. 

## 4. Gut Microbiota and Multiple Myeloma

Increasing evidence suggests an important role of microbes and microbial functions in cancer, ranging from carcinogenesis and progression to response to cancer therapy. With recent studies demonstrating the influence of the gut microbiome on hematological cancers, the correlation between gut microbiota and MM, together with their impact on responses to MM therapy, will be discussed herein (Table 1).

### 4.1. Gut Microbiota Composition and MM

Concerning increasingly developed methods for profiling (16S rRNA sequencing and metagenomic shotgun sequencing), the diversity, along with compositional differences in the gut microbiota, have been studied in MM individuals; this is characterized by a lower diversity of microbiota [26,57] and potential enrichment of opportunistic pathogenic bacteria. Specifically, MM patients had significant enrichment of *Raoultella ornithinolytica*, *Enterobacter cloacae*, *Citrobacter freundii*, *Klebsiella pneumoniae,* etc., in the fecal sample [26]. Another cohort of MM patients also demonstrated significant differences at the stages of healthy individuals and MGUS and MM patients. MM patients had a higher abundance of *Odoribacter* and *Lactobacillus* and a lower abundance of *Blautia* and *Faecalibacterium* than healthy patients; MM patients also had a higher abundance of *Kluyvera* and *Bacteroides* and a lower abundance of *Blautia* and *Parabacteroides* compared with MGUS patients [58]. Furthermore, these dominant forces in the development of MM will be reviewed.

### 4.2. Gut Microbiota and MM Progression

Once the diversity and compositional differences in the gut microbiota have been identified, the next focus would be how they affect MM. Unlike the composition of the gut microbiota between MGUS and MM patients, it has been suggested that the imbalance of gut microbiota may be related to the development of MGUS into MM [59]. In addition, the influence of gut microbiota on hematopoiesis has been explored, demonstrating that Rag-1 deficient mice had a different composition of gut microbiota compared with wild-type mice, and transferable fecal bacteria from wild-type mice may significantly increase hematopoiesis [60]. This suggests that gut microbiota may play an important role in myelomagenesis. Meanwhile, a more direct correlation was studied by Calcinotto et al. [61], which showed that the level of IL-17 in the bone marrow of VK*MYC mice increased at the MGUS stage, and was then closely related to the progression of smoldering MM to MM, suggesting that the higher levels of IL-17 in bone marrow increase the susceptibility of smoldering MM to MM. Moreover, the IL-17 may be related to the stimulation of Th17 cells by gut microbiota. However, the effects of Th17 and its product IL-17 on MM cells are not yet clear. It has been reported that IL-17 promotes the progression of clonal plasma cells, but another study alleviates the progression of MM [62,63].

Furthermore, additional studies have aimed at understanding the gut microbiota and MM progression. In our work, a higher abundance of *Enterobacteriaceae* and *Streptococcaceae* was positively correlated with the ISS stage of MM patients, and enrichment of *Klebsiella pneumoniae* was confirmed to promote MM progression, underlying the effects of the microbiota on MM progression [26]. Meanwhile, intestinal *Prevotella heparinolytica* was reported to accelerate disease progression by promoting intestinal Th17 cell differentiation and migration to bone marrow in MM [61].

### 4.3. Gut Microbiota and MM Therapy

Clinically, the most significant advancements in MM therapy have been the introduction of proteasome inhibitors (bortezomib and carfilzomib), immunomodulatory agents, monoclonal antibodies directed against MM cell surface antigens, and autologous HSCT [2], which significantly prolonged patients’ survival time. However, resistance to therapy and the chances of recurrence are still inevitable. The emerging knowledge of the ability of the gut microbiota to modulate the response to anti-cancer therapy offers new possibilities to improve anti-myeloma efficacy.

#### 4.3.1. Gut Microbiota and Chemotherapy

Huang et al. [64] have discovered that the usage of corticosteroids may cause a change in the composition of gut microbiota, demonstrating a higher abundance of *Bifidobacterium* and *Lactobacillus* and a lower abundance of *Mucispirillum* in mice exposed to corticosteroids. The exposure to dexamethasone also led to substantial shifts in gut microbiota. Additionally, gut microbiota could shape responses to effective treatment with conventional chemotherapy. The effects of cyclophosphamide were abrogated in the antibiotic-treated mice, and the presence of particular bacterial species could increase the sensitivity to cyclophosphamide [65]. In addition to the efficacy of anti-cancer therapy, adverse effects regulated by gut microbiota have also been studied. SCFAs produced by intestinal microbes were reported to alleviate gastrointestinal toxicity caused by proteasome inhibitors [66].

#### 4.3.2. Gut Microbiota and Hematopoietic Stem Cell Transplantation

High-dose chemotherapy combined with auto-HSCT has been considered an important treatment for MM patients [67]. Treated with concurrent therapies that significantly alter the composition of the gut microbiota, including immunosuppressants, broad-spectrum antibiotics, etc, these patients commonly had intestinal dysbiosis [68,69]. Indeed, analyses of longitudinal fecal samples demonstrated a loss of bacterial diversity after HSCT [70,71,72,73,74], demonstrating that the diversity of the gut microbiota of MM patients with auto-HSCT lowered before transplantation, and then lowered further during transplantation [71]. Moreover, MM patients with a low diversity of gut microbiota during transplantation had an impaired transplantation response [72], while those with a high diversity of gut microbiota had better transplantation results and lower transplant-related mortality [71]. In addition, the influence of the gut microbiota on the toxicity to HSCT therapies was investigated. During the pretreatment stage of melphalan, a higher abundance of *Bacteroides* was discovered to be associated with the reduced incidence of severe diarrhea, while a higher abundance of *Blautia* and *Ruminococcus* was closely related to severe diarrhea and post-transplant nausea and vomiting [73]. Meanwhile, the significant susceptibility to bloodstream infections (BSI) remained another toxicity of HSCT with lowered diversity and intestinal dysbiosis [74]. Studies have reported that with low diversity of gut microbiota and dominance of specific bacteria (mainly *enterococcus*, *streptococcus*, and various *proteobacteria*), bacteremia remarkedly increased during HSCT [75].

#### 4.3.3. Gut Microbiota and CAR-T Therapy

Recently, chimeric antigen receptor (CAR) T-cell therapy has changed the therapeutic landscape in MM. Studies have revealed the potential role of gut microbiota in the efficacy and toxicity of CAR-T therapy for B-cell malignancies [76,77]. Hu et al. have found that a higher abundance of *Sutterella* was associated with a complete response and prolonged survival in MM patients with BCMA-CART treatment. The study has also defined the higher abundance of *Bifidobacterium*, *Leuconostoc*, *Stenotrophomonas*, and *Staphylococcus* in patients with severe cytokine release syndrome (CRS), which is reproducible in an independent cohort of 38 MM patients [78], suggesting that gut microbiota play a role in regulating efficacy and CRS of BCMA-CART therapy. However, there is still a great deal to learn more about the gut microbiota and CAR-T therapy, from the inherent mechanisms to optimal strategies to enhance therapeutic responses.

### 4.4. Gut Microbiota and Therapeutic Response

Based on the evidence of the association between gut microbiota and clinical outcomes of MM [71,73], efforts have focused on identifying predictors of the therapeutic response, which reveals that the higher abundance of butyrate-producing bacteria, such as *Eubacterium hallii* and *Faecalibacterium prausnitzii*, associated with negative minimal residual disease (MRD) [79], and together with fecal butyrate, are related to persistent negative MRD [80]. Thus, butyrate-producing bacteria may play a significant role in the evaluation of MRD status and prediction of efficacy in MM patients.

Summarily, with the change in the diversity and composition of the gut microbiota, insights have been gained into the influence of the gut microbiota on MM, not only from the aspects of myelomagenesis and progression but also with regard to the response and toxicity of therapy. Thus, valuable information on how gut microbiota affects MM has drawn the attention of researchers.

**Table 1 cancers-15-01942-t001:** Studies of the gut microbiome in multiple myeloma.

Study (Year)	Patient Population	Microbiome Technique	Alteration of Microbiome	The Differential Microbiota	Findings
Gut microbiota and MM progression					
Jian X. 2020 [26]	Healthy: *n* = 18 MM patients: *n* = 19	Metagenomic shotgun sequencing /qPCR	The lower diversity in MM patients compared to the healthy controls	The significantly lower abundance of Clostridium butyricum and *Anaerostipes hadrus* in MM; The significantly higher abundance of *Klebsiella pneumoniae*, *Citrobacter freundii*, *Streptococcus pneumoniae*, etc, in MM patients.	The SCFA-producing bacteria alleviate MM progression; Nitrogen-recycling bacteria promote MM progression by glutamine; The abundance of nitrogen-recycling bacteria is positively associated with the ISS stage.
Zhang B. 2019 [57]	Healthy: *n* = 17 MM patients: *n* = 61	16S rRNA /qPCR	The lower diversity in MM patients compared to the healthy	At the phylum level: the higher abundance of *Proteobacteria* but the lower abundance of *Actinobacteria* in MM patients; At the genus level: the higher abundance of *Bacteroides*, *Faecalibacterium*, and *Roseburia* in MM patients; Species level: the enrichment of *Pseudomonas aeruginosa* and *Faecalibacterium* in MM.	The abundance of *Faecalibacterium prausnitzii* is significantly correlated with the ISS stage.
Calcinotto A. 2018 [61]	MM mice	16S rRNA	NA	The significantly higher abundance of *Prevotella heparinolytica* in MM mice.	*Prevotella heparinolytica* promotes MM progression by driving Th17 cells.
Gut microbiota and therapeutic response					
Huang EY. 2015 [64]	Mice	16S rRNA	NA	Exposure to dexamethasone, the abundance of *Bifidobacterium* and *Lactobacillus* increased, and the abundance of *Mucispirillum* decreased.	The changes in gut microbiota induced by dexamethasone may affect the therapeutic activity of dexamethasone.
Viaud S. 2013 [65]	Mice	16S rRNA /qPCR	The disruption of the intestinal barrier and alteration of composition by CTX treatment	A reduction of bacterial species of the *Firmicutes* phylum including *Clostridium cluster XIVa*, *Roseburia*, unclassified *Lachnospiraceae, Coprococcus* induced by CTX.	The gut microbiota help shape the anticancer immune response.
Pianko MJ. 2019 [79]	MM patients: *n* = 34	16S rRNA	No significant difference in the α diversity between MRD^-^ and MRD^+^ patients	The increased abundance of *Eubacterium hallii* in MRD^-^ MM patients.	The potential association of *Eubacterium hallii* with treatment response in MM patients.
Gut microbiota and HSCT in MM					
Khan N. 2021 [71]	MM with auto-HSCT *n* = 272	16S rRNA	The lower diversity of the early pretransplant period and decreasing further during transplantation	The increased abundance of Enterococcus, Streptococcus, and Lactobacillus.	Above-median fecal intestinal diversity in the peri-engraftment period is associated with decreased risk of death or progression.
D’Angelo C. 2023 [72]	MM with auto-HSCT *n* = 30	16S rRNA	A significant loss of diversity following transplant	A significant bloom of *Bacteroides* species at engraftment.	The decreased diversity of gut microbiota during transplantation is associated with disease progression.
El-Jurdi N. 2019 [73]	MM with auto-HSCT *n* = 15	16S rRNA	The decrease in diversity, and the recovery within 1 month after transplantation	The abundance of *Bacteroides* and *Blautia* decreased and recovered to the pre-transplantation level one month after transplantation; *Prevotella* and *Streptococcus* decrease during transplantation.	Microbiome composition present at baseline is associated with the incidence and severity of post-transplantation nausea, vomiting, and culture-negative neutropenic fever, as well as with the rate of neutrophil engraftment.
Taur Y. 2012 [75]	MM with allo-HSCT *n* = 8	16S rRNA	The reduction of diversity during allo-HSCT	Enterococcus, Streptococcus, and Proteobacteria increase gradually during HSCT.	During allo-HSCT, the diversity and stability of the gut microbiota are disrupted, resulting in domination by bacteria associated with subsequent bacteremia.
Hu Y. 2022 [78]	RRMM *n* = 43	16S rRNA	A significant decrease in diversity after the CAR-T therapy	The abundance of *Bifidobacterium*, *Prevotella*, *Sutterella*, and *Collinsella* are different in CR and PR patients; Severe CRS presents with a higher abundance of *Bifidobacterium*, *Leuconostoc*, *Stenotrophomonas*, and *Staphylococcus;* Butyricicoccus are enriched in patients with mild CRS.	Therapeutic response in MM and occurrence of severe CRS in MM are associated with specific gut microbiome alterations.

Abbreviation: NA, not applicable; MM, multiple myeloma; SCFA, short chain fatty acid; MRD, minimal residual disease; CTX, cyclophosphamide; auto-HCT, autologous hematopoietic cell transplantation; allo-HCT, allogeneic hematopoietic cell transplantation; RRMM, relapsed/refractory multiple myeloma; BSI, bloodstream infection; CR, complete remission; PR, partial remission; CRS, cytokine release syndrome; BCMA-CART, B-cell maturation antigen (BCMA) chimeric antigen receptor T (CART)-cell therapy.

## 5. Gut Microbiota-Host Amino Acid Metabolism Interaction on Multiple Myeloma

As reviewed above, the gut microbiota is becoming increasingly recognized for its influence on host metabolism. Meanwhile, gut microbiota contributes to MM, not only via the progressive activities of specific bacteria but also through the influence of the microbiota on host metabolism. In particular, the application of high throughput multi-omics analysis to the study of MM (Figure 2) brought a deeper understanding of the association of gut microbiota and host metabolism. Therefore, we further review evidence for gut microbiota-host amino acid metabolism interaction in the progression of MM (Figure 3). 

### 5.1. Gut Microbiota-Host Glutamine Metabolism Interaction on MM Progression

The crosstalk between microbiota and host metabolism is critical. Microbes encode related genes to metabolize dietary nutrients in the gut [24] and produce varieties of metabolites. Among them, L-glutamine (Gln) is one of the microbial metabolites, which is also synthesized from glutamate (Glu) and ammonium (NH_4_^+^) by glutaminase [81] to meet the requirements of both energy generation and as a source of carbon and nitrogen for biomass accumulation in physiological conditions [82]. The maintenance of high levels of glutamine has also been shown to provide a ready source of carbon and nitrogen to drive tumor growth [82,83,84]. As for MM, glutamine has been proven to be important in the growth of cells. Furthermore, researchers have revealed that the gut dominant bacteria were significantly correlated with the serum differential metabolites in MM. Specifically, *Klebsiella pneumoniae* promoted MM progression via de novo synthesis of glutamine [26]. Then, addicted to Gln, MM cells metabolized Gln, leading to an accumulation of NH_4_^+^ in the bone marrow and subsequent release into circulation [26,81,85]. Vice versa, abnormal accumulation of NH_4_^+^ promoted the proliferation of nitrogen-source circulating bacteria, such as *Klebsiella pneumoniae*, thereby promoting the synthesis of glutamine [26]. The data presently available suggest that the crosstalk between gut microbiota and host glutamine metabolism in the MM progression, giving hints that manipulation of gut microbiota–glutamine interaction could be a novel treatment strategy. Another study has focused on pneumonia in MM patients, which accounts for a significant cause of morbidity and mortality [86], demonstrating that the dominant bacteria of *Klebsiella pneumoniae* synthesized glutamine to promote the expression of pulmonary inflammatory factors, and thereby contribute to pneumonia in MM [52].

### 5.2. Gut Microbiota-Host Short-Chain Fatty Acids Metabolism Interaction on MM Progression

A better understanding of the roles of microbes in SCFAs metabolism provides an opportunity to explore gut microbiota-SCFA metabolism interaction in MM. As the most common bacterial end-product involved in amino acid metabolism [87], SCFAs exert beneficial biological activities on MM. Researchers have reported a lower abundance of SCFAs-producing bacteria in MM compared with healthy controls [26,88,89]. The alleviation of progression in MM mice with oral gavage of *Clostridium butyricum* has also been observed [26]. Nonetheless, the inherent mechanisms of SCFAs on MM progression remain poorly understood, SCFAs were reported to skew the balance of inflammatory cytokines locally and systemically or modulate immunity regarding cancer development and progression or response to therapy [90,91]. In fact, the functions of the anti-tumor role have been extended to recognize mechanisms that alleviate MM progression, including SCFAs that either inhibit inflammatory factors (such as IL-1β, IL-6, and TNF-a) or induce the expression of anti-inflammatory factors (such as IL-10), all of which were considered to be at a high level in the bone marrow microenvironment [74].

These studies have highlighted a role in the interaction between gut microbiota and host amino acid metabolism (mainly glutamine and SCFAs) in the MM progression. Furthermore, additional studies on the interaction of gut microbiota and host amino acid metabolism in MM remain to be defined.

### 5.3. Clinical Translational Insights of the Microbiota-Host Amino Acid Metabolism Interactions

Given the above presentation, the clinical translations of the interactions have attracted the attention of scientists. Based on the MM-enriched microbiota, host metabolites, such as creatinine, aminomalonic acid, and L-proline, were reported to be positively associated with *Enterobacter cloacae*, *Klebsiella pneumoniae*, and *Klebsiella variicola*; their abundance was positively reflected with ISS stages in MM patients [26]. There are still limited studies that examine the interaction between gut microbiota and host amino acid metabolism in MM. As the field moves forward, there may be renewed interest in these clinical translations.

Clinical translations made by modulating the gut microbiota may represent a novel adjunct of modalities of MM patients. The prophylactic antibiotics were presumed to alleviate MM progression or improve therapeutic efficacy by intervening in the gut microbiota to eliminate pathogenic bacteria. However, after the first demonstration that administration of antibiotics abrogated anti-tumor activity in a murine model of melanoma [92], the TEAMM results showed similar results, which demonstrated the higher mortality of MM patients with 12-week treatment of prophylactic levofloxacin [93]. Subsequent studies drew the same conclusion, that an imbalance of gut microbiota caused by prophylactic antibiotic treatment may promote MM progression [94,95]. In addition, more relevant reports showed that disrupting the gut microbiota by antibiotic use could impair the anti-tumor efficacy [65,96,97,98].

Fecal bacteria transplantation (FMT) is considered the most direct and effective means to manipulate gut microbiota. To our knowledge, various factors, including conditioning regimens, infections, antibiotics use, and immune response, lead to drastic and rapid perturbations of the gut microbiota during HSCT. FMT has been proven to be an optimal modulation for patients with HSCT to restore the diversity of gut microbiota [99,100,101,102], as well as an effective strategy for HSCT-associated *Clostridium difficile* infection [103,104] and steroid-resistant acute GVHD [105,106]. With the potential transmission of antibiotic-resistant pathogens, engineered microbial transplantation is another approach in the preclinical stage. In addition, intervening with dietary and pre-/probiotic supplementation is also of important consideration to improve the outcomes of cancer [107]. From this aspect, a study has reported that MM patients with healthier prediagnosis dietary habits may have longer survival than those with less healthy diets [108], suggesting the potentiality of the dietary intervention for MM.

## 6. The Challenges of Gut Microbiota in Multiple Myeloma: Future Directions

### 6.1. Limitations of Microbiome Analysis and Studies

The profiling of microbiome and metabolome has been increasingly yielding new information on the altered microbiota and metabolites by the utility of omics. However, with the complexities and the unknowns of microbiota, the concern of optimal sequence methods and the responding analytical methods (such as 16S rRNA sequencing or metagenomic shotgun sequencing, the choice of which analytical databases) still exist, especially for the interpretation of analytic results that will be translated into clinical applications. In addition, the microbiota imbalance, which influences the phenotype of the observed mice, may not be entirely attributable to bacteria, but may involve the other microbiota (fungi, viruses, etc.) [109,110], hinting at the need for cautious interpretation in clinical translations. Then, inconsistency existed in different studies. Studies on the altered microbiome and metabolome of MM patients have been identified. However, without enough consistency across different studies and considerations of differences in host interaction with environment and diet, which may vary substantially, the limitation of cross-study comparison inevitably exists. Furthermore, recognizing the specific bacteria involved, as they vary by disease and patient population, can create a better fundamental understanding of the heterogeneity of MM as well as an ability to predict the effects of gut microbiota-host amino acid metabolism interaction on MM; as of now, however, this topic is poorly understood. Finally, regarding the interaction between gut microbiota and host amino acid metabolism, there is not yet a great deal of literature to draw more general conclusions, and it is unclear why this interaction contributes to MM (especially myelomagenesis). Overall, although many challenges remain, building a better understanding of the roles of microbes and metabolites may enable a powerful new tool for improving the outcomes of MM.

### 6.2. Challenges of the Microbiome in Future Clinical Translations

Most current studies focusing on the modulation of gut microbiota have been in mice, and therefore, the translation of these achievements to clinical applications remains challenging. First, the differences between mice and humans should not be ignored. The similarities between mice are amplified for several reasons, such as relative homogeneity (identical genetic backgrounds, diet, and similar environment), and the substantial cage effects [111]. Conversely, the many inherent inter-subject variabilities, especially for disease activity and medication use, lead to biological outcomes that have obscure meanings. Second, to identify the correlation of different bacterial species with a clinical response by FMT, these humanized mice mainly reproduce the bacterial diversity of the donor microbiota [112,113]. However, these mice are not identical to humans in the aspect of physiological and immune responses [114]. Indeed, studies have demonstrated the differences [115,116]. Finally, FMT has been shown to successfully benefit patients in some diseases. However, the process of FMT is difficult and expensive; it also has the potential to transfer to other diseases. These factors necessitate careful considerations of donor selection, conditioning regimen, and banking for potential future autologous transplant.

### 6.3. Future Directions

Preclinical studies on the role of gut microbiota and host metabolism in MM have brought us to focus on this potentially dominant mediator in MM and therapy. However, there is still a great deal to learn concerning the inherent mechanisms, predictive significance, and efficacy to modulate the interaction of gut microbiota and host metabolism, probably ranging from definitive cause–effect relationships, screening of biomarkers in the aspect of diagnosis and dynamic surveillance, to the selection of modulation. In the process, the standardized approaches of collection and analyses of specimens, together with the integration of available data, will bring more valuable information and strategies to target MM therapy. Especially for the modulation, clinical trials targeting this approach for MM patients are in development. Importantly, multicenter studies are critical to minimize geographic differences. Although many challenges remain, building a better understanding of the roles of microbes and metabolites may enable a powerful new tool for improving the outcomes of MM.

## 7. Conclusions

There is compelling evidence that the gut microbiota affects host amino acid metabolism in MM, and that manipulation of microbiota may augment response to anti-myeloma therapy in preclinical models. Based on the influence of gut microbiota on host amino acid metabolism, the abnormal amino acid metabolism, as well as the altered microbiome in MM have been first defined, and these items influence MM progression and the therapeutic response. Then, the gut-microbiota–host-amino-acid metabolism interaction (mainly glutamine and SCFAs) in the MM progression, together with modulation of the gut microbiota (such as FMT) as a novel adjunct regimen of MM patients, has been highlighted. Overall, only through a comprehensive understanding of these interactions can we learn to optimally modulate the gut microbiota to enhance the clinical outcomes of MM.

## Figures and Tables

**Figure 1 cancers-15-01942-f001:**
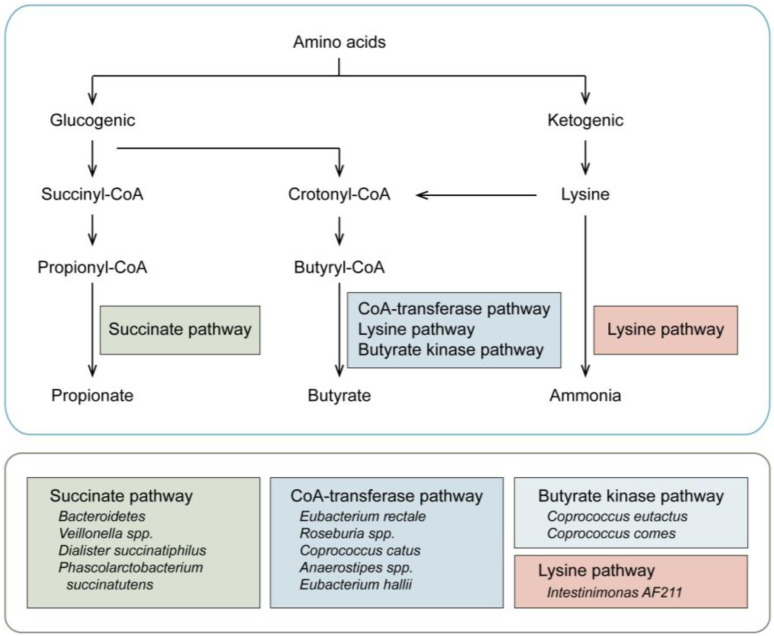
Key metabolic pathways of amino acids in the gut. Through the function of fermentation, the ketogenic and glucogenic metabolic pathways are processed to produce ammonia and SCFAs (butyrate and propionate). In detail, as for the ketogenic process, the amino acids are converted to lysine, and some bacteria then cause the production of ammonia (Lysine pathway). In addition, lysine is metabolized to butyryl-CoA for the production of butyrate (CoA-transferase, succinate, and butyrate kinase pathway). Concerning the glucogenic pathway, the microbiota is mainly for the final synthesis of butyrate, and the production of propionate by the succinate pathway.

**Figure 2 cancers-15-01942-f002:**
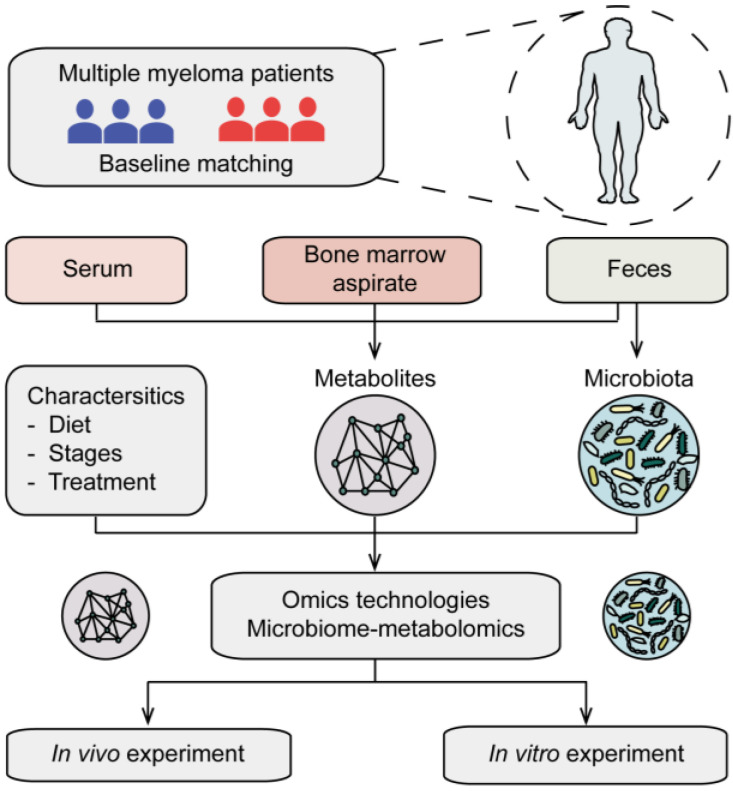
Application of microbiome/metabolomics in multiple myeloma. Eligible MM cohorts with matched baselines are recruited for sampling, then collected along with clinical characteristics for further analysis. Samples of serum, bone marrow aspirate and feces are performed with metabolomics, while paired fecal samples are utilized for microbiome sequencing. Furthermore, an analysis of differential metabolites and microbiota associated with MM is conducted, and then, in vivo and in vitro experiments are performed for further verification and mechanistic study.

**Figure 3 cancers-15-01942-f003:**
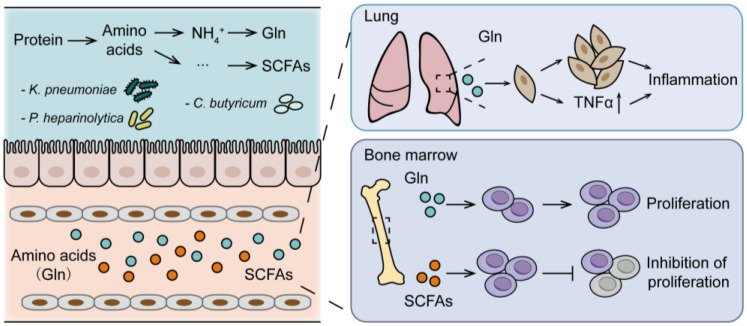
Interaction of gut microbiota and host amino acid metabolism in multiple myeloma. The amino acids are converted to ammonia. Conversely, ammonia is re-utilized by nitrogen bacteria to produce glutamine, wherein the bone marrow, glutamine, which is addicted to MM cells, is utilized by MM cells for proliferation. When it accumulates in the lung, glutamine contributes to the proliferation of lung normal fibroblast cells and elevated secretion of TNF-α for inflammatory infiltration. In addition, the SCFAs produced in the gut are adsorbed and distributed in the bone marrow, alleviating the proliferation of MM cells.

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
