# Peer review of "The Interaction between Gut Microbiota and Host Amino Acids Metabolism in Multiple Myeloma"

_cancers, 2023, doi:10.3390/cancers15071942_

Round 1

Reviewer 1 Report

Yang et al in the review titled `The interaction between gut microbiota and host amino acids metabolism in multiple myeloma1 describe the connection and interaction between microbiota, metabolism and cancer, in their case multiple myeloma.

The review is well written and ideas clearly organized, figures and tables are of a good quality.

The article describes a very important subject, the modulation of microbiota in patients with MM, considering that MM is the second most common hematological malignancy and that there is a great resistance to therapy.

They are also making a parallel with other types of cancer and cancer in general. The authors also stress the importance of decreased variety of gut microbiota in MM and how that affects MM progression, host amino acid metabolism and therapy response.

This review could serve as a link to other studies of gut microbiota and MM treatment.

Author Response

Dear Reviewer:

Thank you for your comments concerning our manuscript entitled "The interaction between gut microbiota and host amino acids metabolism in multiple myeloma" (Manuscript ID: cancers-2278052). Thank you for your positive and insightful comments. The comment is valuable and very helpful for guiding our research.

Reviewer 2 Report

The overall concept of this manuscript titled, “The interaction between gut microbiota and host amino acids metabolism in multiple myeloma” by Yang et al shows light on an interesting topic. I have some concerns regarding the following:

1.        The authors should highlight on the clinical aspect of the microbiota-host amino acid metabolism interaction

2.       The authors need to introspect on the impact of TCA metabolites interaction between gut microbiota and host in MM

3.       For potential future readers the authors should discuss the shortcomings of their work and future translational aspect in a separate paragraph.

4.       It is best if the conclusions paragraph is splitted from future directions

Author Response

Dear Reviewer:

Thank you for your comments concerning our manuscript entitled "The interaction between gut microbiota and host amino acids metabolism in multiple myeloma" (Manuscript ID: cancers-2278052). Those comments are all valuable and very helpful for improving the quality of our review. We have carefully studied all the comments and revised our original manuscript with red marks in it. We hope the revision will meet the publication standard. The responses to the reviewers’ comments are listed as follows:

Reviewer #2 (Remarks to the Author):

The overall concept of this manuscript titled, “The interaction between gut microbiota and host amino acids metabolism in multiple myeloma” by Yang et al shows light on an interesting topic. I have some concerns regarding the following:

Comment 1: The authors should highlight on the clinical aspect of the microbiota-host amino acid metabolism interaction.

Response to comment 1: 

We appreciate your valuable comments. A detailed description was inserted in the 5.3 section, which was also presented as follows.

5.3 Clinical translational insights of the microbiota-host amino acid metabolism interactions

Given the above presentation, the clinical translations of the interactions attract the attention of scientists. Based on the MM enriched microbiota, host metabolites, such as creatinine, aminomalonic acid, and L-proline, were reported to be positively associated with Enterobacter cloacae, Klebsiella pneumoniae, and Klebsiella variicola, of which their abundances were positively reflected with ISS stages in MM patients. Until now, there are still limited studies in the field of interaction between gut microbiota and host amino acid metabolism in MM. As the field moves forward, renewed interest may be taken into consideration in the translations.

Clinical translations by modulating the gut microbiota may represent a novel adjunct of modalities of MM patients. The prophylactic antibiotics were presumed to alleviate MM progression or improve therapeutic efficacy by intervening in the gut microbiota to eliminate pathogenic bacteria. Nevertheless, after the first demonstration that administration of antibiotics abrogated anti-tumor activity in a murine model of melanoma, TEAMM results showed similar results, which demonstrated the higher mortality of MM patients with 12-week treatment of prophylactic levofloxacin. Subsequent studies drew the same conclusion that an imbalance of gut microbiota caused by prophylactic antibiotic treatment may promote MM progression. In addition, more relevant reports showed that disrupting the gut microbiota by antibiotic use could impair the anti-tumor efficacy.

Fecal bacteria transplantation (FMT) is considered the most direct and effective means to manipulate gut microbiota. To our knowledge, various factors, including conditioning regimens, infections, antibiotics use, and immune response, lead to drastic and rapid perturbations of the gut microbiota during HSCT. FMT has been proven to be an optimal modulation for patients with HSCT to restore the diversity of gut microbiota, as well as an effective strategy for HSCT-associated Clostridium difficile infection and steroid-resistant acute GVHD. With the potential transmission of antibiotic-resistant pathogens, engineered microbial transplantation is another approach in the preclinical stage. In addition, intervening with dietary and pre-/probiotic supplementation is also of important consideration to improve the outcomes of cancer. From this aspect, a study has reported that MM patients with healthier prediagnosis dietary habits may have longer survival than those with less healthy diets, suggesting the potentiality of the dietary intervention for MM.

Comment 2: The authors need to introspect on the impact of TCA metabolites interaction between gut microbiota and host in MM.

Response to comment 2: 

We appreciate this constructive suggestion. The field of the impact of TCA metabolites interaction between gut microbiota and host in MM is profound for a better understanding of MM. Until now, the studies mainly aimed at the end-products of amino acids, while the role of secondary metabolites on MM during the metabolic process of amino acids is still unclear, thereby hardly reviewing and drawing general conclusions. Importantly, optimal approaches and preliminary analysis of the unanswered questions may be addressed in future research. Moreover, future deeper understandings may bring this topic to light as a potential mechanism and modulation in MM.

This field provides researchers with future potential research directions. This meaningful hint was inserted into our manuscript marked in red.  

Although our understanding remains somewhat lacking, we have summarized the host pyruvate metabolism in gut microbiota, which is demonstrated as follows. As reviewed, pyruvate can either be catabolized into lactate or generate succinate and acetyl-CoA through the process of the TCA cycle. Nevertheless, these intermediates are detected without a high concentration in typical fecal samples. In fact, they are then metabolized to the SCFAs (acetate, propionate, and butyrate), which are recognized as the most abundant and well-studied microbial end-products. Additionally, small amounts of alcohol can be generated as end-products by pyruvate fermentation. Among them, studies have highlighted a role for SCFAs in MM, which have been reviewed in our manuscript. Generally, SCFA produced by intestinal microbes was reported to alleviate gastrointestinal toxicity caused by proteasome inhibitors. Moreover, SCFAs exert beneficial biological activities on MM, albeit the inherent mechanisms of SCFAs on MM progression remain poorly understood. Evidence from current studies has demonstrated the potential influence on the balance of inflammatory cytokines locally and systemically, and the regulation of immunity.

Comment 3: For potential future readers, the authors should discuss the shortcomings of their work and future translational aspect in a separate paragraph.

Response to comment 3: 

Thank you for pointing it out. The challenges of gut microbiota in MM have been emphasized as a separate section, containing limitations of microbiome analysis and studies, challenges of the microbiome in future clinical translations, and future directions, all of which were marked in red and presented as follows.

  1. The challenges of gut microbiota in multiple myeloma: future directions

6.1 Limitations of microbiome analysis and studies

The profiling of microbiome and metabolome has been increasingly yielding new information on the altered microbiota and metabolites by the utility of omics. However, with the complexities and the unknown of microbiota, the concern of optimal sequenced methods and responding analytical methods (such as 16S rRNA sequencing or metagenomic shotgun sequencing, the choice of which analytical databases) still existed, especially for the interpretation of analytic results to translate into clinical applications. In addition, the microbiota imbalance, which influences the phenotype of the observed mice, may not be entirely attributable to bacteria but may involve the other microbiota (fungi, viruses, etc.), hinting at the cautious interpretation in the clinical translations. Then, inconsistency existed in different studies. Studies on the altered microbiome and metabolome of MM patients have been identified. However, without enough consistency across different studies and considerations of differences in host interaction with environment and diet, which may vary substantially, the limitation of cross-study comparison exists inevitably. Furthermore, the recognition of what specific bacteria vary by disease and patient population, thereby enhancing a better fundamental understanding of the heterogeneity of MM, and an ability to predict the effects of gut microbiota-host amino acid metabolism interaction on MM, remains poorly understood. Finally, as for the interaction between gut microbiota and host amino acid metabolism, there isn’t yet a great deal of literature to draw more general conclusions, and it is still unclear why they contribute to MM (especially myelomagenesis). Overall, although many challenges remain, building a better understanding of the roles of microbes and metabolites may enable a powerful new tool for improving the outcomes of MM.  

6.2 Challenges of the microbiome in future clinical translations

Most current studies focusing on the modulation of gut microbiota have been in mice, and therefore, the translation of these achievements to the clinic remains a challenge. Firstly, the differences between mice and humans should not be ignored. Probably for relative homogeneity (identical genetic backgrounds, diet, and similar environment), together with the substantial cage effects, the similarities between mice are amplified with fewer differences. Conversely, with lots of inherent variabilities, especially for disease activity and medication use, inter-subject variability exists, therefore leading to obscure meaningful biological outcomes. Secondly, to identify the correlation of different bacterial species with a clinical response by FMT, these humanized mice mainly reproduce the bacterial diversity of the donor microbiota. However, these mice are not identical to humans in the aspect of physiological and immune responses. Indeed, studies have demonstrated the differences. Additionally, FMT has been shown to successfully benefit patients in some diseases. But the process of FMT is difficult and expensive, together with the potential to transfer other diseases, all of which necessitates careful considerations of donor selection, conditioning regimen, and banking for potential future autologous transplant.

6.3 Future directions

Preclinical studies on the role of gut microbiota and host metabolism in MM have brought us to focus on this potentially dominant mediator in MM and therapy. However, there is still a great deal to learn concerning the inherent mechanisms, predictive significance, and efficacy to modulate the interaction of gut microbiota and host metabolism, probably ranging from definitive cause-effect relationships, screening of biomarkers in the aspect of diagnosis and dynamic surveillance, to the selection of modulation. In the process, the standardized approaches of collection and analyses of specimens, together with the integration of available data, will bring more valuable information and strategies to target MM therapy. Especially for the modulation, clinical trials targeting this approach for MM patients are in development. Importantly, multicenter studies are critical to minimize geographic differences. Although many challenges remain, building a better understanding of the roles of microbes and metabolites may enable a powerful new tool for improving the outcomes of MM.

Comment 4: It is best if the conclusions paragraph is splitted from future directions

Response to comment 4: 

Thank you for this comment. We have already split the conclusion paragraph from future directions, which indeed made our review more clear and clarified.

Reviewer 3 Report

Yang and colleagues presented in the review, the relationship between gut microbiota, metabolism, and multiple myeloma together with strategies to modulate the microbiota, and some ongoing and future research.

The presented topic is highly relevant and interesting since the effect of the intestinal microbiota probably plays a greater role than was thought a short time ago. It is therefore important to focus attention on this area.

The introduction of the review is well-written, easy to understand, and underlined with recent literature. 

In Table 1, the studies listed dealing with the gut metabolome in multiple myeloma.

The authors nicely presented in several chapters the interaction of the gut microbiota and host amino acid metabolism in multiple myeloma. The review gives a comprehensive insight into the topic. The readers can follow up on the literature if they need more detailed information.

After reading the review, I can only recommend accepting the manuscript in its current form.

Minor:

- Fig. 3 - spelling error "Inflammation" 

Author Response

Dear Reviewer:

Thank you for your comments concerning our manuscript entitled "The interaction between gut microbiota and host amino acids metabolism in multiple myeloma" (Manuscript ID: cancers-2278052). Those comments are all valuable and very helpful for improving the quality of our review. We have carefully studied all the comments and revised our original manuscript with red marks in it. We hope the revision will meet the publication standard. The responses to the reviewers’ comments are listed as follows:

Comment: Minor: - Fig. 3 - spelling error "Inflammation"

Response to comment:

We greatly appreciate your positive comments. Specifically, we have corrected the spelling error. Moreover, we have confirmed all linguistic expressions.

Round 2

Reviewer 2 Report

Authors have clarified all my concerns